# Nanocomposites Based on Antiferroelectric Liquid Crystal (S)-MHPOBC Doping with Au Nanoparticles

**DOI:** 10.3390/molecules27123663

**Published:** 2022-06-07

**Authors:** Sebastian Lalik, Olaf Stefańczyk, Dorota Dardas, Aleksandra Deptuch, Tetiana Yevchenko, Shin-ichi Ohkoshi, Monika Marzec

**Affiliations:** 1Institute of Physics, Jagiellonian University, 30-348 Kraków, Poland; sebastian.lalik@doctoral.uj.edu.pl; 2Department of Chemistry, School of Science, The University of Tokyo, Tokyo 113-0033, Japan; olaf@chem.s.u-tokyo.ac.jp (O.S.); ohkoshi@chem.s.u-tokyo.ac.jp (S.-i.O.); 3Institute of Molecular Physics, Polish Academy of Sciences, 60-179 Poznań, Poland; dardas@ifmpan.poznan.pl (D.D.); tetiana.yevchenko@ifmpan.poznan.pl (T.Y.); 4Institute of Nuclear Physics, Polish Academy of Sciences, 31-342 Kraków, Poland; aleksandra.deptuch@ifj.edu.pl

**Keywords:** antiferroelectric liquid crystal, Au nanoparticles, metal–organic composites, localized surface plasmon resonance

## Abstract

Modification of the physical properties of the (S)-MHPOBC antiferroelectric liquid crystal (AFLC) by doping with low concentrations of gold nanoparticles is presented for the first time. We used several complementary experimental methods to determine the effect of Au nanoparticles on AFLC in the metal–organic composites. It was found that the dopant inhibits the matrix crystallization process and modifies the phase transitions temperatures and switching time, as well as increases the helical pitch and spontaneous polarization, while the tilt angle slightly changes. We also showed that both the LC matrix and Au nanoparticles show strong fluorescence in the green light range, and the contact angle depends on the temperature and dopant concentration.

## 1. Introduction

The process of doping soft matter with various types of nanomaterials, such as nanoparticles, nanorods, and quantum dots, is a fairly convenient alternative to designing new materials without a typical chemical approach. This way is technically simpler, economically cheaper, and what is the most important, it gives very promising effects: modification of the parameters of a soft matter matrix (liquid crystal, and polymer). It should also be noted here that a more sophisticated approach is the liquid crystal gold nanoparticles (NPs) synthesis, in which the mesogenic unit is chemically bonded to the NP’s surface. This approach produces materials with a high Au content, as opposed to doping, where agglomeration occurs at higher concentrations [1,2,3,4,5,6,7,8,9,10,11,12,13]. Hybrid materials such as liquid crystal + nanomaterials have been the subject of research for the last few years [14,15,16]. Since nematic liquid crystals (NLCs) are the most widely used in the industry, great interest is focused on them as a matrix for the production of nanocomposites. However, it seems that more complex phases such as the ferroelectric SmC* or the antiferroelectric SmC*_A_ can also be modified in the same way. Our research and this paper are an attempt to fill in the gap, in which the antiferroelectric SmC*_A_ phase is an organic matrix doped with Au nanoparticles. We have chosen gold nanoparticles, because they are very often used as a dopant, and their synthesis and properties are well-described in the literature [17,18]. Below, we provide a brief overview of the research of nanocomposites based on liquid crystals and Au nanoparticles to demonstrate the need for our research and explain our motivation.

Considering calamitic non-chiral nematic liquid crystals, it was shown that, after doping with Au nanoparticles (decorated with different surfactants), depending on the liquid crystal (LC), the photoluminescence (PL) increases, the red shift appears for the absorption band as a result of localized surface plasmon resonance (LSPR), the nematic-isotropic phase transition temperature and threshold voltage decrease, and the Fréedericksz voltage for both H = 0 and H > 0 increase [19,20,21,22]. It was also found that the relaxation frequency decreases after doping, as explained by the generation of a local electric field in the nanocomposite [23]. In another case, the absorption increased after doping with Au NPs, and a blue shift was noted for LSPR [24,25]. On the other hand, by doping polymer dispersed LC (PDLC) with Au nanoparticles, the significant increase of the relative light transmission was noted and explained by the increase in a local electric field by excitation of the nanoparticles’ surface plasmons [26]. Tiwari et al. gave direct evidence for the presence of surface plasmons in NLC doped with Au NPs [27]. A liquid crystal elastomer exhibiting the N phase was also doped with Au nanoparticles by Montazami et al., who showed the increase of the nematic–isotropic phase transition temperature and stiffness of the nanocomposite (Young modulus increases) [28]. Likewise, Au nanoparticles have been used as a dopant to NLCs from the cyanobiphenyls homologous series (nCB). Depending on the homolog of the defect-free layers, increases in the dielectric anisotropy Δε, the optical anisotropy Δn, elastic constant K, viscosity γ, and broadening of the N phase were found [29,30,31,32,33]. The modifications of the material parameters were explained as a results of creating clusters of two different types: (a) nanoparticle clusters with LC molecules (for smaller concentrations) and (b) nanoparticle clusters among themselves (for higher concentrations) [34]. In turn, the red shift of LSPR after doping was explained by the anchoring of LC molecules to the Au surface [35]. For the binary mixtures 5CB:6CB (various concentrations) doped with Au NPs, a large decrease in the viscosity γ, threshold voltage, and τ_off_ was found [36]. An admixture of Au NPs also caused an increase of the specific electric conductivity in the N and Is phases and broadening of the N and SmA phases [37,38,39]. Au nanoparticles have been also used to align NLC molecules. For example, Urbański et al. achieved homeotropic ordering in planar cells, which is presumably due to Au NP accumulation at the interface polymer/NLC and opposes the polymer effects [40], while Hegmann et al. found that Au NPs with hydrophobic ligands introduced into NLC can form birefringent stripe domains separated by homeotropically aligned LC molecules [41,42]. In turn, Yoshida et al. stabilized the blue phase by trapping nanoparticles in disclination lines, which lowered the empty volume and free energy [43].

The discotic liquid crystals (DLC) have also been used as a matrix for doping with Au nanoparticles, and depending on the liquid crystal used as the host matrix, it was found, for example, that after doping the decrease of the Col_h_–Is phase transition temperature, an increase of the optical energy gap and an increase of the conductivity was observed [44,45].

A few papers presented the influence of gold nanoparticles on ferroelectric liquid crystals (FLC). For example, after doping a decrease of the Goldstone mode’s relaxation frequency and the SmA*–SmC* phase transition temperature, an increase of the conductivity was found [46]. Additionally, after doping with Au NPs, an increase of the photoluminescence (PL) intensity without changing the qualitative PL spectrum, the memory effect and an increase of the tilt angle were observed [15,16,47]. Fluorescent LC materials are still a rare group, because the shapes and structures of many fluorescent moieties are not optimal for mesophase formation [48]. FLCs were also doped with Au nanorods, similar to how it was done by Podgornov et al. and Shukla et al. [49,50].

As shown above, mainly nematic liquid crystals have been used as a matrix for doping with Au nanoparticles, although discotic and polymeric and even ferroelectric liquid crystals have also been used. However, no antiferroelectric liquid crystals were doped. Therefore, we prepared two composites based on the well-known AFLC—(S)-MHPOBC doped with various concentrations of Au nanoparticles: Composite 2 (0.2 wt. % Au NPs) and Composite 3 (0.5 wt. % Au NPs) and compare their properties with the undoped sample—Composite 1 (0.0 wt. % Au NPs) by using the following complementary methods: polarizing optical microscopy, differential scanning calorimetry, X-ray diffraction, fluorescent confocal polarizing microscopy, ultraviolet–visible spectroscopy, optical/scanning electron microscopy, contact angle, and electro-optic measurements. It has been shown how many AFLC parameters can be changed by a small amount of Au nanoparticles in a surprising and intriguing manner. We interpret the results of the research on new hybrid materials on various planes, determining the influence of the admixture on both the phase sequence and the helix pitch, fluorescence, morphology, and wettability of the layers, as well as electro-optic parameters (switching time, tilt angle, and spontaneous polarization). We use the nomenclature recommended by the IUPAC—in particular, the phase composed of chiral molecules is denoted by adding an asterisk to the name of the phase [51].

## 2. Results and Discussion

### 2.1. Results of Polarizing Optical Microscopy (POM) Method

By using the polarizing microscopy method, textures for the SmA*, SmC*_α_ (which is often noted as ferrielectric one, and other designations SmC*_1/3_ = SmC*_FI1_), SmC*, SmC*_γ_, SmC*_A_, SmI*, and SmI* + Cr_1_ phases were registered with decreasing temperatures for Composites 1–3. As an example, the textures for Composite 3 are presented in Figure 1, while all the Composites are in Appendix A. It is seen that all the Composites studied exhibit a rich phase polymorphism, which is usually observed for optically pure (S)-MHPOBC or (R)-MHPOBC [52]. In the SmA* phase, director n (also represents the optical axis in this uniaxial phase) is oriented along the smectic layer normal k [51].

On cooling from the isotropic liquid, the SmA* phase starts to form as anisotropic, elongated ovules (commonly called bâtonnets) for all Composites. Finally, the observed textures for the SmA* phase in all Composites contain topological focal conic defects leading to focal conic fan textures in the SmA* phase for Composites 1–3. The smectic layers are arranged in Dupin’s cyclides, which include a pair of focal conics [53]. The considered focal conic defects are usually common when the molecules anchor strongly to the cell surface or other nucleation points (e.g., an admixture or spacers) and the defects appear in random places on the texture (Appendix A) [54]. No effect of the Au nanoparticles was observed on both fans and focal conic defects in the texture of the SmA* phase.

The SmC*_α_ sub-phase appeared immediately below the SmA* phase and is usually very narrow (1 to 2 °C), although there are some reports about a wider SmC*_α_ sub-phase (ca. 3 °C) in the analogs of (S)-MHPOBC [55,56,57]. Both the SmA* and SmC*_α_ phases are uniaxial, with good approximation (optical axis is along the normal layer). However, the SmC*_α_ phase shows a helical structure with tilted molecules, but the tilt angle is much smaller than in the SmC* phase, and the helix pitch is usually of the order of several smectic layers [53]. As is seen in Figure 1 and Appendix A, though the texture of the SmC*_α_ phase is slightly different from the texture of the SmA* phase (the visible ripples formation), the difference between textures of the SmC*_α_ and SmC* phases was difficult to find; they were similar, and the distinction between these phases was based on a subtle color change and more visible striation along the fan regions. Usually, the focal conic defects degeneration in the form of their breaking and separation is observed at the transition to the SmC*_γ_ phase. However, it was not observed by us for Composites 1–3. In our case, the SmC*_γ_ phase was identified on the basis of the appearance of strong striations across the fans regions [51]. During further cooling down, the SmC*_A_ was recognized by a slight color change. Focal conic defects persist in the phase transition and are present throughout the SmC*_A_ phase. Moreover, the stripes appear in the low temperature range of this phase (Figure 1 and Appendix A). The transition between the SmC*_A_ and SmI* antiferroelectric phases is well-visible as the color of the texture changes. Dierking also reported about striations in the SmC*_A_, as well as in the SmI*, phases for pure MHPOBC [53]. At a low temperature range, the SmI* phase, the nucleation of the crystallization is visible (Figure 1 and Appendix A), which results in the coexistence of these two phases over a wide temperature range, until full crystallization occurs. Interestingly, the crystallization process (nucleation and crystallization propagation) was much faster in a pure matrix (Composite 1) than in Composites 2 and 3. For the pure matrix, the Cr_1_ phase can be observed practically already at 40 °C (Figure 1h). This proves that the admixture inhibits the crystallization process.

Summarizing, the addition of Au nanoparticles did not change the texture of all the phases identified in the Composites studied. Therefore, one can conclude that an admixture has no visible effect on the molecular ordering (e.g., the focal conic defects). Focal conic defects disappear only during the crystallization process. At the present stage, it is difficult to conclude whether the admixture modifies the nucleation or the crystallization propagation. However, due to the article’s nature, we do not focus on the crystallization process.

### 2.2. Results of Differential Scanning Calorimetry (DSC) Method

The calorimetric curves with three strong anomalies were recorded during heating for Composites 1–3 (Figure 2). Additionally, the anomalies corresponding to the following phase transitions: SmC*_A_–SmC*_γ_, SmC*_γ_–SmC*, and SmC*_α_–SmA* are visible in the insets in Figure 2a. The SmC*–SmC*_α_ phase transition is not visible, although it was observed by Chandani et al. for (R)-MHPOBC at a very low heating rate [58]. It is related to the rate of temperature changes. In our case, the heating/cooling rate was equal to 10 °C/min, and therefore, only one anomaly corresponding to the SmC*–SmC*_α_–SmA* phase transition is visible. Moreover, four crystal phases are visible, which has not been previously reported for pure (S)-MHPOBC. Interestingly, additional phases were registered within the crystal region for Composite 3 (with the highest Au nanoparticles amount) (Figure 2b,c). During cooling, the same phase transitions were registered, except the transitions in the crystal region (see Appendix A). Appendix A presents the influence of Au nanoparticles on the transition temperatures of the Iso–SmA*, SmC*_A_–SmI*, and SmI*–Cr_1_ phase transitions. The Iso–SmA* phase transition for Composite 2 is decreased by 0.3 °C as compared to Composites 1 and 3, while the SmI*–Cr_1_ phase transition for Composite 3 is increased by 0.4 °C as compared to Composites 1 and 2. In turn, the SmC*_A_–SmI* phase transition temperature is not disturbed by the admixture. To sum up, the admixture of the Au nanoparticles slightly and differently influences the particular phase transitions during cooling and heating as well. The individual phase transition temperatures were determined based on the so-called onset T_o_ or peak T_p_ temperatures. The latter is marked with an asterisk in the phase sequences (only applies to transitions around 120 °C due to the difficulty of determining T_o_) (Table 1).

It was found that the enthalpy change ΔH decreases with the increasing concentration for the Iso–SmA* phase transition during cooling, as was also observed by Tripathi et al. for other liquid crystalline compounds [59]. In turn, the total ΔH connected with the phase transitions between sub-phases during cooling does not change, within the uncertainty limit, after doping.

**Table 1 molecules-27-03663-t001:** Phase sequences during heating and cooling for Composites 1–3 obtained by texture observations and the DSC method. The asterisk next to temperature denotes the peak temperature T_p_, ΔT = ± 0.1 °C.

**Heating**	**Composite 1**
	**Cr_1_ (53.1 °C) Cr_2_ (68.8 °C) Cr_3_ (73.7 °C) Cr_4_ (81.8 °C) SmC*_A_ (*119.1 °C) SmC*_γ_ (*119.9 °C) SmC*/SmC*_α_ (*122.0 °C) SmA* (148.5 °C) Iso**
	**(S)-MHPOBC—Fernandes et al. [60]**
	**SmC*_A_ (118.2 °C) SmC*_γ_ (119.2 °C) SmC* (120.7 °C) SmC*_α_ (*122.0 °C) SmA***
	**Composite 2**
	**Cr_1_ (52.5 °C) Cr_2_ (68.9 °C) Cr_3_ (73.7 °C) Cr_4_ (81.9 °C) SmC*_A_ (*119.1 °C) SmC*_γ_ (*119.9 °C) SmC*/SmC*_α_ (*122.0 °C) SmA* (148.4 °C) Iso**
	**Composite 3**
	**Cr_1_ (60.0 °C) Cr_2_ (65.3 °C) Cr_2_’ (70.5 °C) Cr_3_ (73.8 °C) Cr_4_ (81.9 °C) SmC*_A_ (*119.9 °C) SmC*_γ_ (*120.7 °C) SmC*/SmC*_α_ (*122.4 °C) SmA* (148.9 °C) Iso**
**Cooling**	**Composite 1**
	**Iso (149.4 °C) SmA* (122.0 °C) SmC*_α_/SmC* (*119.5 °C) SmC*_γ_ (*118.1 °C) SmC*_A_ (65.0 °C) SmI* (27.3 °C) Cr_1_**
	**Composite 2**
	**Iso (149.1 °C) SmA* (121.8 °C) SmC*_α_/SmC* (*119.5 °C) SmC*_γ_ (*117.6 °C) SmC*_A_ (63.1 °C) SmI* (27.2 °C) Cr_1_**
	**Composite 3**
	**Iso (149.4 °C) SmA* (122.3 °C) SmC*_α_/SmC* (*119.8 °C) SmC*_γ_ (*118.2 °C) SmC*_A_ (64.8 °C) SmI* (27.6 °C) Cr_1_**

The enthalpy change ΔH connected with the chosen phase transitions during heating is presented in Figure 2d. The greatest impact of the Au NPs on the enthalpy changes was found for the Cr_4_–SmC*_A_ phase transition in Composite 3 (an increase of ca. 26J/g). Hence, even a small concentration of Au NPs (0.2 and 0.5 wt. %) strongly influences the transition between the crystal and liquid crystalline phases. Such an increase in the enthalpy change is inextricably linked with an increase of the entropy change ΔS in Composites 2 and 3. It cannot rule out that the surfactant-decorated Au nanoparticles (with a chemical structure similar to the end fragments of (S)-MHPOBC molecules) interact in a specific way with the LC molecules and are responsible for the energetic hindering of this phase transition. It is seen that ΔH after doping decreases in Composite 2 but increases in Composite 3 in the SmA*–Iso phase transition. The slight increase in ΔH for the Cr_2_–Cr_3_ phase transition was registered for Composite 2. Concluding, the Au NPs admixture affects more the transition parameters within the crystal than the liquid crystalline phases during heating.

Fernandes et al. determined the phase transition temperatures for (S)-MHPOBC by using differential scanning calorimetry with a 0.2 °C/min heating rate [60], and they are also presented in Table 1 for comparison. It seems that transition temperatures obtained by us for Composite 1 are in good agreement with those obtained by Fernandes et al., except that based on the onset temperature T_o_ (SmC*_A_–SmC*_γ_ and SmC*_γ_–SmC*). Additionally, Fernandes et al. showed that, for the high optical purity of MHPOBC, the ferrielectric SmC*_β_ sub-phase exists instead of the ferroelectric SmC* phase (which appears for a lower purity of MHPOBC), while all sub-phases can exist in the intermediate optical purity compound [60]. According to the manufacturer, the chemical purity of our (S)-MHPOBC is 99%, but the optical purity is not specified. Taking the above into account, we assumed that we observed the ferroelectric SmC* phase in the LC matrix. It should also be mentioned here that the temperature range of the SmI* phase in (R)-MHPOBC strongly decreased with the decreasing cooling rate (from ca. 35 °C at 2.0 °C/min to disappearance at 0.5 °C/min), while the SmC*_A_–SmI* phase transition temperature was not dependent on the cooling rate [61]. We recorded the calorimetric curves with a fairly high heating/cooling rate equal to 10 °C/min; therefore, two distinct phase transitions were registered on the temperature scale: SmC*_A_–SmI* and SmI*–Cr_1_. In turn, as we show below, in the case of TLI measurements, the time domain measurement is longer and the crystallization occurs at much higher temperatures, which essentially resulted in the coexistence of the SmI* and Cr_1_ phases. Such coexistence was registered by the polarizing microscope method at a 2 °C/min rate, as presented in Figure 1 and Appendix A.

### 2.3. Results of Transmitted Light Intensity (TLI) Method

Due to the difficulty in determining by DSC and texture observation methods the SmC*–SmC*_α_ and SmC*_α_–SmC* phase transition temperatures, the complementary method, i.e., TLI, was used. The transmitted light intensities versus temperature registered during cooling for Composites 1–3 are presented in Figure 3a, while Figure 3b,c present the enlarged temperature ranges 116–124 °C and 40–70 °C, respectively. As is seen, both the SmA*–SmC*_α_ and SmC*_α_–SmC* phase transitions are visible as subtle changes in the intensities (small structural changes between these phases and sub-phase) for all Composites 1–3 studied. During further decreasing of the temperature at a rate of 2 °C/min, the transmitted light intensity increases down to the middle of the SmC*_A_ phase by about 30%. Within the SmC*_A_ phase, a parabolic shape for the TLI signal is observed for each sample. This phenomenon is probably connected with layer thickness changes, as it was revealed by the X-ray diffraction method, and will be presented below. A relatively small but distinct change in TLI dependence is visible at ca. 52 and 54 °C for Composite 1 and Composites 2 and 3, respectively (Figure 3c), which could be attributed to the initiation of the SmI*–Cr_1_ phase transition. Indeed, the SmI* phase begins to coexist with the emerging Cr_1_ phase, as confirmed by the inclusion of the Cr_1_ phase in the SmI* phase visible in the textures (Figure 1). The formation moment of the first nucleation points is related to the discontinuities in Figure 3c marked by dashed arrows. Below the discontinuity is the range of the two-phase coexistence. On the other hand, an anomaly around 26 °C associated with SmI*–Cr_1_ phase transition is visible in the DSC curves. It is most likely related to the different cooling rates in the various methods (10 °C/min in DSC and 2 °C/min in TLI). The SmC*_α_–SmC* phase transition temperatures with the initiation temperature for the coexistence of the SmI*–Cr_1_ phases in Composites 1–3 are presented in Figure 3d. The phase coexistence of SmI*–Cr_1_ was not detected by the DSC method (the anomaly connected with the SmI*–Cr_1_ transition is narrow), which is probably connected to a higher cooling rate in the DSC method.

On the other hand, the Cr_1_-Cr_2_, Cr_4_–SmC*_A_, and SmA*-Iso phase transitions are very well visible for all studied composites during heating (Appendix A). Interestingly, within the SmC*_A_ phase, a monotonic decrease in the light intensity with increasing temperature is visible (in contrast to a parabolic shape for cooling). In turn, the subtle SmC*–SmC*_α_ phase transition is visible in the enlargement presented in Appendix A.

### 2.4. Results of X-ray Diffraction (XRD) Method

Representative diffraction patterns for the Composite 3 collected on cooling are shown in Figure 4a. The patterns of the SmA* and SmC*_A_ phases are similar, therefore the sub-phases SmC*_α_, SmC*_γ,_ and the SmC* phase existing in a narrow temperature range cannot be distinguished [61]. The hexatic SmI* phase of (S)-MHPOBC is not observed in diffraction results due to slow cooling [61]—crystallization starts before the SmC*_A_–SmI* phase transition is reached. The position of the low-angle peak at 2θ ≈ 2.6° is related to the smectic layer spacing (smectic layer thickness) D by the Bragg equation [62]. The temperature dependence of D is presented in Figure 4b. The layer thickness D is practically the same for all Composites 1–3 and equal to 34.5–34.8Å in the SmA* phase (increases slowly on cooling) while in the SmC*_A_ phase, it decreases on cooling down to 33.4–33.6 Å at 95–100 °C, therefore the relative change of D in all smectic phases is ca. 4%. However, the increase of D on cooling below 95 °C (and decreasing of TLI signal at the same time) is an indicator of a subsequent hexatic phase, similarly to what was observed for other compounds [63,64]. Similar smectic layer spacing D was reported by Inui et al. for AFLC (TFMHPDOPB) while smaller (D_max_~31Å) for fluorinated orthoconic compound was reported by Milewska et al. [65,66]. Summarizing, both the transmitted light intensity and XRD methods confirm the existence of a hexatic phase below the anticlinic SmC*_A_ one as was observed by the polarizing microscopy method.

The crystallization is indicated by arising of numerous sharp diffraction peaks, which are observed for all samples at 68–70 °C. The low-angle peak characteristic for a crystal phase is observed at 2θ ≈ 2.4°, which corresponds to the distance between the crystal planes (37.0–37.2Å), constant with the decreasing temperature down to −10 °C.

The wide diffuse maximum at 2θ ≈ 20° in the diffraction patterns of the isotropic liquid and smectic phases arises from the short-range positional order. In the scattering vector space (q = 4πsinθ/λ) the maximum has a form of a Lorentz distribution, I(q) = (1 + ξ^2^(q−q_0_)^2^)^−1^. The correlation length of the short-range order ξ, as well as its temperature dependence, is almost the same for studied Composites 1–3 except for the Composite 2 in the SmA* phase where little decrease with temperature is visible (Figure 4c). The fast increase of ξ below 80 °C is caused by approaching the hexatic SmI* phase [63]. The diffraction results show that the admixture of Au nanoparticles does not affect significantly the structural parameters of the smectic and the crystal phases for (S)-MHPOBC. Additionally, the crystallization process is not disturbed by Au admixture.

### 2.5. Results of Fluorescence Confocal Polarizing Microscopy (FCPM) Method

The fluorescence images in subsequent slices of the samples registered in the spectral range 490–590 nm in SmC*_A_ phase (at 68 °C) are presented in Figure 5 while for selected temperature in this spectral range as well as in 430–455 nm are presented in Appendix A. Fluorescence was observed in both wavelength ranges for all Composites 1–3, however with different intensities. Interestingly, for Composites 2 and 3 the decrease in fluorescence intensity is visible in the 490–590 nm range with decreasing temperature in the SmC*_A_ phase (68, 88 °C) as well as at higher temperatures for Composite 3. The places with more intense color at a particular slice of the Composites 1–3 can mean: a greater accumulation of Au nanoparticles (see Appendix A) or structure defects (which is especially good visible for green fluorescence for all Composites 1–3 at 118 °C and 122 °C, Appendix A). Au nanoparticles may accumulate in structural defects to minimize the free energy of the system, which causes an increase in fluorescence in these places. In turn, pure Au nanoparticles fluorescence is visible in the green (490–590 nm) range while it is not observed in blue (430–455 nm) as is seen in Figure 5 and Appendix A. We can state that this fluorescence is coming from Au nanoparticles because the surfactant decorated Au NPs (1-octanethiol) does not have fluorescence due to the lack of appropriate chromophores fragment.

To sum up, from the qualitative point of view, the doping (S)-MHPOBC with Au nanoparticles affects the fluorescence intensity depending on the temperature (the phase). This can be a valuable application determinant for a given hybrid material. We have also analyzed the fluorescence images numerically (see Appendix A). It turned out that both in the green and blue ranges, there was an increase in the fluorescence intensity after doping with Au NPs and the increase is greater in the green light range.

### 2.6. Results of Ultraviolet–Visible (UV–Vis) Spectroscopy—Free Droplet Method

The temperature dependence of the helix pitch determined for Composites 1–3 is presented in Figure 6. The maximum absorbance along with the uncertainty was appointed based on fitting the Gaussian curve to the experimental data and the helix pitch values p = λ_s_/2n_av_ in the SmC* and p = λ_s_/n_av_ in the SmC*_A_ phases were calculated (the average refractive index n_av_ = 1.5 for the SmC* and SmC*_A_ phases) [67,68]. The helix pitch in the SmC*_γ_ phase was calculated as for the antiferroelectric SmC*_A_ phase. A similar situation was for the SmI* phase which is typically antiferroelectric. The helix pitch at the SmC*_γ_–SmC*_A_ phase transition is equal to 599.4, 603.5, and 661.2 nm for Composites 1, 2, and 3, respectively, and down to ca. 90 °C is almost constant. Below 90 °C for Composites 2 and 3, the helix pitch decreases with the decreasing temperature down to ca. 70 °C, where it slightly increases. It is most likely related to the SmC*_A_–SmI* phase transition. In turn, for Composite 1 ((S)-MHPOBC) a very subtle increase in the helix pitch from about 90 °C is visible. In the SmI* phase, the helix pitch is almost the same within the uncertainty for Composites 1and 2 while it is larger about 48 nm for Composite 3.

However, the situation is different in the sub-phases and the SmC*_A_ phase. Already in the SmC*_γ_ phase, an increase in helix pitch is visible after the addition of Au nanoparticles (by approx. 5 nm for Composite 2 and approx. 62 nm for Composite 3). The value of the helix pitch for Composite 1 ((S)-MHPOBC) is similar as that observed for (R)-MHPOBC by Chandani et al. [58]. A decrease of the helix pitch with decreasing temperature is visible for Composites 2 and 3 in the SmC*_A_ phase, in particular for Composite 3 from about 95 °C, which may be due to an increase in viscosity and a decrease in the mobility of Au NPs with decreasing temperature. In turn, in the low temperature range of the SmC*_A_ phase, the helix pitch is practically the same for Composites 1 and 2, while, for Composite 3, it is about 45 nm longer. The matrix at low temperatures may act as a self-assembling medium for the nanoparticles. The agglomeration process (the overlapping of thiol chains of one nanoparticle with the another) at such a concentration (0.5 wt. %) should also be taken into account and may be responsible for the increase of helical pitch in Composite 3. It should be remembered the Au nanoparticles of 2–4 nm diameter are the same size as the smectic layer thickness (the length of the LC molecule). Thus, such an object disturbs the order of a large number of molecules producing structural defects as well as may be responsible for shifting the entire layer in the space, which will affect the helical pitch. Therefore, the increase in helical pitch after doping in Composites 2 and 3 is due to the presence of Au NPs and is greater for Composite 3 due to the presence of more Au NPs in Composite 3 than Composite 2. The greater number of Au nanoparticles in Composite 3 means that more of them will be located between smectic layers, so the helix pitch will be increased more. Taking into account the results of XRD, i.e., no influence of Au nanoparticles on the thickness of the smectic layers, it seems that decorated Au nanoparticles in the (S)-MHPOBC matrix accumulates mainly between the smectic layers, increasing the helix pitch. It is also possible that some of the Au nanoparticles will be located in the smectic layers, which should be visible in the change of the tilt angle of the molecules and will be discussed later in the paper.

### 2.7. Results of Optical Microscopy (OM)/Scanning Electron Microscopy (SEM) Methods

High-resolution digital optical images were registered for two different types of illumination (ring and coaxial) for Au nanoparticles and pristine Composites as well as Au nanoparticles and Composites cooled to room temperature after being heated to 155 °C and kept at this temperature for 5 min (Figure 7 and Figure 8 and Appendix A). The color of pristine Composite 1 was white while pristine Composites 2 and 3 were light pink, because the Au nanoparticles are pink (see Figure 7 and Figure 8). It is seen that the color intensity correlates with the increase in the concentration of Au nanoparticles—it is pinker for Composite 3 with a higher concentration of Au nanoparticles. For pristine Au nanoparticles at 50× and 100× magnification, the network of white lines on the pink area is visible (Figure 7, upper row). In our opinion, these lines are microscale cracks that resulted from the evaporation of the solvent during sample preparation. At 400× and 2000× magnification, chaotically arranged pillars with a length of less than 30 μm are visible in the sub-micro scale. What is interesting, after heating to 155 °C, annealing at this temperature for 5 min, and cooling down to room temperature the color changes to purple and microscale lines are also visible while the pillar’s structure is not (Figure 7, bottom row). In our opinion, the pillars visible at room temperature are Au nanoparticles which self-assemble through the chains of surfactant during isothermal evaporation of the solvent (toluene). In turn, heating to 155 °C causes the destruction of these structures due to thermal reorientation of the 1-octanethiol chains. The break of strong Au-S chemical bonds is also possible under the influence of temperature while the transition of surfactant is ruled out because the melting and boiling point of 1-octanethiol are −49 °C and 199 °C, respectively.

**Figure 7 molecules-27-03663-f007:**
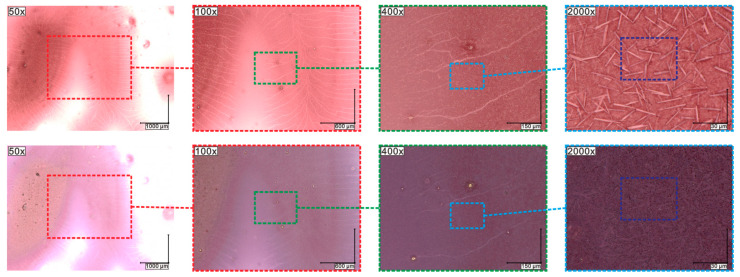
Room temperature digital optical images with ring illumination for pristine Au nanoparticles (**upper row**) and cooled to room temperature after being heated to 155 °C and kept at this temperature for 5 min (**bottom row**). The colored dotted squares represent the enlarged areas.

High-resolution digital optical images registered with coaxial illumination for Composites 1–3 at room temperature are presented in Appendix A while for cooled to room temperature after being heated to 155 °C and kept at this temperature for 5 min in Appendix A. As is seen in Appendix A there is no influence of Au nanoparticles (pillars are not visible) on the images of Composites 2 and 3, they are the same as for Composite 1. Probably, the use of sonication during the preparation of Composites 2 and 3 did not allow the formation of a lamellar structure, visible for pristine Au nanoparticles (prepared without sonication). On the other hand, for Composites 2 and 3, after heating and then cooling to room temperature, dark objects are visible, and invisible before heating at 155 °C (Figure 8, Appendix A). For example, for Composite 3, for a magnification of 2000×, dark objects appear in random places, not observed for pristine Composite 3, and for a magnification of 6000×, a lot of white dots additionally appear (Figure 8). The same is true for Composite 2, while these objects do not appear for Composite 1 (Appendix A). The number of these objects increases with the content of Au nanoparticles—it is higher for Composite 3 (Appendix A). In our opinion, annealing of Composites 2 and 3 at 155 °C (above the clearing temperature of (S)-MHPOBC) causes the movement of Au nanoparticles in the liquid crystal matrix (in the isotropic phase) which leads to their agglomeration, visible as dark objects (big aggregates) for a magnification of 2000× and white spots (small aggregates) for 6000×. It should be noted here that before heating Au nanoparticles were well dispersed in the LC matrix and these objects were not visible. In summary, the annealing of pure Au nanoparticles changes their color from pink to purple while the annealing of Composites leads to the melting of the (S)-MHPOBC matrix and the formation of Au nanoparticle agglomerates of various sizes (e.g., white dots approx. 500 nm in diameter).

**Figure 8 molecules-27-03663-f008:**
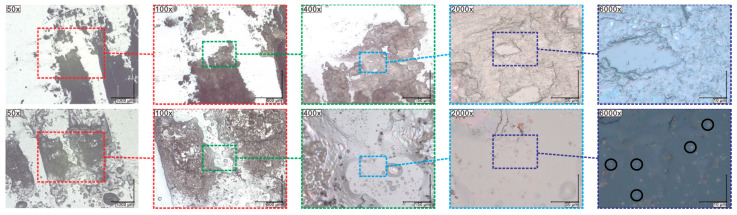
Room temperature digital optical images with coaxial illumination for those pristine (**the first row**) and cooled to room temperature after being heated to 155 °C and kept at this temperature for 5 min in Composite 3 (**second row**). The colored dotted squares represent the enlarged areas.

In SEM images at 50× magnification, pristine (S)-MHPOBC forms many crystals while Composites 2 and 3 form larger flakes (Appendix A). For the remaining magnifications, the images are similar for all Composites, there is no visible influence of Au nanoparticles. For the 100× and 400× magnifications, flat terraces with a large area without any special features while for a magnification of 2000× clear needle structures (representing clusters of matrix chains) are visible. On the other hand, SEM images of pristine Au nanoparticles, even at 50× and 100× magnification, show some irregular morphological structures, and at 400× magnification, a very rough surface with numerous small anisotropic structures is visible. However, for 2000× and 6000×, fairly flat terraces appear between these structures. These anisotropic structures appear to be large clusters decorated with Au nanoparticles that stick out above the terrace surface, thereby contributing to the roughness of the surface.

The real colors of the studied materials, obtained by superimposing the SEM images on the optical images, are shown in Figure 9. After evaporation of the solvent (toluene), Au nanoparticles have a strongly pink color, while the (S)-MHPOBC crystals become more and more intensely pink as the concentration of Au nanoparticles grows. As the intensity of the pink color at each image location for Composites 2 and 3 (Figure 9b,c) seems more or less the same, it means that the structures visible on the micrometer scale (terraces) contain well dispersed Au nanoparticles.

### 2.8. Results of Contact Angle (CA) Measurements

The study of the wettability of the Composites was carried out as a function of temperature. Exemplary water droplets deposited on the studied thin layers of Composites 1–3 and silicon wafer (reference) are shown in Appendix A. The contact angle Θ_CA_ of a pure silicon wafer (as a reference) is temperature independent and equal to ca. 55° in the studied temperature range (Figure 10a). On the other hand, the contact angle for Composites 1–3 in the range from 0 °C to ca. 50 °C is constant and equal to ca. 90° for Composites 1 and 3 and ca. 100° for Composite 2. Such high contact angles indicate poor wettability of the surface of Composites 1–3 by water in this temperature range. From a temperature of about 50 °C, a continuous decrease in the contact angle with increasing temperature is visible, and near the phase transition to the SmC*_A_, in a narrow temperature range, a step is visible (dashed ellipses in Figure 10a), most likely related to the temporary coexistence of crystal and the SmC*_A_ phases. A bigger contact angle (weaker wetting) for the Composite 2 at low temperatures (Figure 10b) may be connected with a relatively larger number of alkyl chains (fragments of the 1-octanethiol) terminated with a methyl group which is exposed on interface composite-air (a methyl group is a hydrophobic group decreasing the wettability). In turn, higher wetting of the surface of Composites 2 and 3 compared to Composite 1 is observed in the temperature range of about 40–80 °C (crystal phases region) because the conformational chains change with increasing temperature which improves the wettability. Moreover, the Θ_CA_ for the Composite 3 in the SmC*_A_ phase is about 15° higher than for the Composites 1 and 2 (Figure 10c) but to explain this further research is needed.

### 2.9. Results of Electro-Optic (EO) Measurements

The effect of Au nanoparticles on the switching time was investigated in the broad temperature range (Figure 11 and Appendix A). In the SmC*_A_ phase (down to ca. 65 °C) the switching time is shorter than 100 μs while below 65 °C its significant increase is visible for all Composites (which is caused by an increase in viscosity). Such behavior is typical for antiferroelectric liquid crystals [57,69]. In the SmC*_A_ phase (up to 95 °C) the switching time is shorter for Composite 3 than for Composite 1 while the longest is for Composite 2 while above 95 °C it is the same for Composites 1 and 3 and ca. 10 μs shorter than for Composite 2. The shorter switching time for Composite 3 relative to Composite 1 may be due to the local field modification (increase) after doping [70]. In our opinion, a shorter switching time for Composite 3 relative to Composite 1 may be due to several reasons. A decrease in the material viscosity after doping, modification of the local field that acts on the molecules, the agglomeration process as well as interactions between LC molecules and Au nanoparticles should be considered [70]. In our case, a shorter switching time for Composite 3 relative to Composite 1 may be due to the local field modification (increase) after doping, as a dominant factor.

The temperature dependence of the tilt angle in the SmC*_A_ for all studied Composites is presented in Figure 12. The characteristic plateau is visible below ca. 85 °C for all the samples. At the lowest temperatures, a decrease in the tilt angle indicates the SmC*_A_–SmI* phase transition. There is no influence of concentration 0.2 wt. % Au nanoparticles on the Θ values (Composite 2) while Θ for Composite 3 is greater than for Composite 1, and this difference decreases as the temperature decreases (e.g., it is ca. 3° at 114 °C). This behavior can be explained by the fact that with an increase in Au NPs concentration: (a) statistically, in more places a single smectic layer is disturbed and more smectic layers are disturbed, (b) 3D structures may form in the bulk by Au nanoparticles self-organization or (c) agglomeration process (although decorated NPs were used) can take place [71]. All of this is directly responsible for modifying the arrangement of LC molecules, and hence for changing the tilt angle—the primary order parameter. 1-octanethiol chains on adjacent Au NPs may interact, which may increase the tilt angle of LC molecules. An increase in Au nanoparticles concentration may cause self-assembly processes in many places of LC matrix ((S)-MHPOBC), which increases the tilt angle Θ. However, taking into account that the layer thickness D(T), helix pitch p(T) and tilt angle Θ(T) for Composites 1 and 2 are practically the same (the helix pitch is only increased by ca. 1 nm), one can assume that the low concentration of Au NPs does not disturb the original liquid crystal order. In turn, for Composite 3 the tilt angle as well as the helix pitch increases while the layer thickness does not change. Therefore, it seems that Au nanoparticles are located both in the layers (modification Θ) and between them (modification p) in Composite 3. The inset in Figure 12 shows a bar diagram with the Θ values for selected temperatures to better visualize the differences between Composite 1 and 3. The tilt angle determined by us is in good agreement with the tilt angle reported for MHPOBC analogous in several papers [72,73,74,75,76,77].

The spontaneous polarization P_s_ was determined in the SmC*_γ_, SmC*_A_, and SmI* phases for all Composites. The results obtained for Composite 1 are in good agreement with the results presented in various papers [73,74,78,79,80] while for Composites 2 and 3 with the results presented in [81]. Figure 13a presents the spontaneous polarization versus applied voltage U_AC_ in the SmC*_A_ (at 117 °C) and as is seen the non-zero P_s_ is observed at 20 V for all Composites. Thus, the Au nanoparticles have no influence on the threshold voltage in this phase. The spontaneous polarization increases with further increasing the applied voltage, but no saturation is visible up to 160 V. The highest P_s_ is for Composite 3 while the lowest for Composite 1.

It is known that the spontaneous polarization depends directly on the dipole moment of the LC molecule therefore its increase may be connected with the better ordering of LC molecules in the Composites 2 and 3 than in Composite 1 or with an effect of localized surface plasmon resonances. The localized plasmon resonance for Au nanoparticles appears in the visible wavelength range [71] and depends on many factors, such as size, shape, dielectric constant, environment, and even the distance between adjacent nanoparticles [82]. The samples of Composites were exposed to white light during the electro-optic measurements which may be responsible for the generation of localized surface plasmon resonance. Localized surface plasmon resonance is known to be associated with the oscillation of electrons on a metallic nanoparticle, excited by light at a frequency corresponding to that of electrons (see Figure 14) [71]. During these oscillations, the electron cloud shifts, and the dipole moment is induced, and consequently, the electric field near the surface of the nanoparticles is strongly amplified. Thus, LC molecules in the vicinity of such nanoparticles will locally sense a greater electric field than the applied external electric field. This enhancement quickly disappears as the distance from the nanoparticle surface increases and the absorption band for Au NPs in the LC matrix can be relatively wide. Due to the relatively large size of nanoparticles (comparable to the length of LC molecules), the plasmonic resonance of a given nanoparticle (enhancement of the electric field) will be felt by the greater number of LC molecules, the higher the concentration of Au in the nanocomposite, as it was observed by us (see Figure 13a). In summarizing, it appears that the LSPR is responsible for the increase in the spontaneous polarization P_s_ for Composites 2 and 3 relative to Composite 1 down to 90 °C (see Figure 13), while below this temperature the spontaneous polarization, within the measurement uncertainty limit, is the same for Composites 1–3. Such a situation at lower temperatures may result, among others, from the increase in matrix viscosity.

It should be also noted here that, during the measurement of spontaneous polarization, around 97 °C the sample response splits into two peaks (the formation of the two helices) and slight wrinkles on the monodomain appear, which are evidence of the SmC*_A_ phase [73].

The increase of spontaneous polarization after doping the ferroelectric liquid crystal with Au NPs was reported by several authors. For example, Malik et al. explained this phenomenon by trapping parasitic ions by dopant while Shukla et al. attributed this to the chirality of decorating surfactant and the kind of material used to functionalize Au NPs [83,84]. In turn, two localized surface plasmon resonances were observed in the FLC mixture (CHS1) doped with Au nanorods: longitudinal (λ ≈ 851 nm) and transverse (λ = 541 nm) [85]. The larger scattering is due to the bigger cross-section. According to Ahmad et al. Au nanoparticles can take colors in the range of orange, red, brown, etc., depending on the core size, and usually show an absorption band in the range of 500–550 nm [86]. The smaller nanoparticles have an absorption maximum of around 520 nm, the larger ones scatter more and the shape of the absorption band is widened with the maximum shifted towards the red light. After aggregation, the conduction electrons near the surface of each nanoparticle are delocalized and divided (shared) by the nearest nanoparticles which change the optical properties by shifting the surface plasmon resonance to lower energies (red shift). Therefore, non-aggregated Au nanoparticles absorb blue light while Au aggregated nanoparticles leads to significant shifts in the frequency of plasmon resonance towards red, broadening the absorption band and changing the solution color [86]. The simulated absorption spectra for Au nanoparticles (20–100 nm) in water turned out that for larger the diameter of nanoparticles, the absorption maximum shifts towards red light and the absorption efficiency initially increases with the size up to diameters equal to 70 nm, and then begins to decrease (radiation scattering begins to dominate over absorption) [87]. The determination of the exact wavelength at which the localized surface plasmon resonance occurs for the Au nanoparticles used requires further research. However, due to the spherically symmetric Au nanoparticles shape, only one absorption band associated with LSPR is expected. The influence of Au nanoparticles concentration on the (S)-MHPOBC absorption spectra will be also studied and presented in the next paper.

## 3. Materials and Methods

### 3.1. Materials

Commercially available liquid crystal (LC) abbreviated as (S)-MHPOBC (Sigma Aldrich Co., Saint Louis, MO, USA) was used as a liquid crystal matrix. Its chemical structure is shown in Figure 15a. The LC was in the form of white flakes and was used without chemical pre-purification. The commercially available 1-octanethiol-functionalized Au nanoparticles (spherical, diameter 2–4 nm) as a 2% *w*/*v* solution in toluene (Sigma Aldrich, Co., Saint Louis, MO, USA, Figure 15b) were used as an admixture. The size of Au nanoparticles was chosen as comparable to the average lengths of LC molecules. Due to the lack of commercial Au nanoparticles without surfactant, it is important to highlight the advantages and disadvantages of the decoration process. It is important because the surfactant chains can be largely responsible for modifying the matrix parameters. The nanoparticle functionalization is aimed at: (a) ensuring and controlling the miscibility of nanoparticles with the matrix (elimination of phase segregation processes) and (b) elimination or strong reduction of the agglomeration process. However, the functionalization procedure also has some disadvantages: (a) inhibition of direct interactions between the nanoparticle and the matrix molecules and (b) additional difficulties in the interpretation of the results for composites due to the presence of an additional chemical reagent—the decorating factor [82].

Three composites were prepared with three different concentrations of Au nanoparticles: Composite 1 (0.0 wt. % Au NPs—pure LC matrix), Composite 2 (0.2 wt. % Au NPs), and Composite 3 (0.5 wt. % Au NPs). We used low weight concentrations to not disturb the liquid crystal order in individual smectic layers. Composites 2 and 3 were prepared according to the following scheme:Appropriate amounts of pure (S)-MHPOBC were weighed and flooded with the appropriate volume of Au nanoparticles solution (see Table 2) and 2 mL of toluene was added.The solutions were left for a few hours (~19 h).The solutions were sonicated for 30 min in the 30(on)/30(off) mode with amplitude A = 40% (in the presence of ice as an external cooling agent).The solutions were placed on the magnetic stirrer (~100 °C, 1400 rpm) for about ~18–19 h to slowly evaporate half of the solvent volume.The sonication process was performed for 15 min in the 45(on)/15(off) mode with A = 40%.The solutions were poured (~50 μL) on cleaned microscope slides and left for solvent evaporation under ambient conditions for over 70 h.The prepared composites were gently scratched from a glass by metal blade and used for measurements.

The above procedure (but without the Au nanoparticles) was used to prepare Composite 1 (pure matrix). The prepared Composites 1–3 differed significantly in color, namely Composite 1 was completely white, Composites 2 and 3 adopted violet-brown colors, and their intensity depended on the concentration of Au NPs (see Figure 16b).

After pouring the solutions on microscope slides and the toluene evaporating, we noticed the matrix is relatively fragile, it detached easily from the slide and took the powder form. However, after adding Au nanoparticles the situation was slightly different, the fragility degree decreased. In our opinion, the cause is the presence of a surfactant (1-octanethiol) rather than the presence of Au nanoparticles.

### 3.2. Polarizing Optical Microscopy and Electro-Optic Measurements

Nikon Eclipse LV100POL polarizing microscope (Nikon, Tokyo, Japan) equipped with Fine Instruments WTMS-14C heating stage (Elektronika Jądrowa, Cracow, Poland) coupled to a Canon EOS600D digital camera (Canon, Tokyo, Japan) and PC was used to determine the phase sequence of studied composites. The complete set for electric measurements included: Agilent 33120A AC voltage generator, F20-FLC Electronics amplifier, Agilent DSO6102A digital oscilloscope, INSTEC PD02 photodetector, tunable resistor system (a 100 kΩ resistor was used for all measurements), Canon EOS600D digital camera (Canon, Tokyo, Japan). Composites 1–3 were placed in ITO electro-optic cells with planar alignment (thickness 4.9 μm, active electrode area 25 mm^2^, AWAT Company, Warsaw, Poland) by capillary effect at a temperature just above the clearing point and then slowly cooled down to room temperature.

The texture observations were carried out at cooling (155–0 °C, cycle I) and heating (20–155 °C, cycle II) at a rate of 2 °C/min. The textures processing was performed using ImageJ (LOCI, University of Wisconsin, Madison, WI, USA).

The light transmitted intensity by the sample was measured with a photodetector. The time-domain signal was averaged to give an intensity value at a specific temperature. Then all results for a given composite were normalized to the range [0, 1]. Measurements were taken both during heating and cooling in 0.5 °C steps with a rate 2 °C/min, there was 1 min before each measurement for temperature stabilization.

Before electro-optic measurements, the sample was heated up to an isotropic liquid and then slowly cooled down to the first phase below SmA* and aligned under an AC electric field (rectangular, 100 Hz, 120 V) for about 6–8 h. Among others, such a long time is due to the low helix pitch value for (S)-MHPOBC [88]. Measurements of the spontaneous polarization, switching time, and tilt angle (120 V_pp_, 24.5 MV/m) were done every 1 °C. The reversal current method was used to measure P_s_ and τ while Θ was measured between Clark-Lagerwall states [89,90,91] by using a photodetector. A triangular signal was used for P_s_ measurements, while a rectangular signal was used for τ and Θ measurements. The numerical calculation was done using the Origin 2018 program (OriginLab, Northampton, UK). The uncertainty related to the P_s_ was equal to 1%. The systematic uncertainties of τ are related to the time oscilloscope base for a specific measurement cycle, while the uncertainty for Θ is a complex uncertainty: systematic uncertainty of 0.1° and statistical uncertainty resulting from the obtained values Θ_1_ and Θ_2_.

### 3.3. Differential Scanning Calorimetry

The phase transition temperatures and their enthalpy changes during heating and cooling were determined by using a PerkinElmer DSC8000 differential scanning calorimeter (PerkinElmer, Waltham, MA, USA). The calorimeter was calibrated to the melting points of water and indium. Samples weighing 7.80 mg of Composites 1–3 were placed in aluminum cups and closed with a press. The measurement was done at a rate of 10 °C/min during heating (0–170 °C, cycle I) and cooling (170–0 °C, cycle II). Results were analyzed using Pyris software (Perkin Elmer, Waltham, MA, USA) and Origin 2020 (OriginLab, Northampton, UK).

### 3.4. X-ray Diffraction

XRD measurements were performed using the CuK_α_ radiation with Empyrean 2 (Malvern Panalytical, Worcestershire, UK) diffractometer with Cryostream 700 Plus (Oxford Cryosystems) temperature attachment. The samples were introduced to capillaries (borosilicate glass, 0.3 mm diameter) by the capillary effect in the isotropic liquid phase. The XRD patterns were collected on cooling in geometry of a horizontal rotating capillary with a parabolic mirror on an incident beam. The measurement for a powder Si (SRM 640b) was performed to determine the zero of the diffractometer and reduce the systematic error in 2θ values. The patterns were analyzed in WinPLOTR [92].

### 3.5. Fluorescence Confocal Polarizing Microscopy

The FCPM method was originally used to study liquid crystals due to the defects present in them. In our case, however, it is the method used to detect fluorescence in Composites 1–3 [93,94].

Olympus Fluoview 1200 fluorescence microscope (Olympus, Tokyo, Japan) with a diode laser (λ D 405 nm and λ D 473 nm) and the beam power equal to 1.24 mW was used. The low beam power value is intentional, to avoid light-induced reorientation of LC molecules. The collected images show the autofluorescence obtained simultaneously for two excitation wavelengths and detection ranges. The autofluorescence signal was detected in the spectral range 430–455 nm (blue) and 490–590 nm (green) in the reflection mode. To obtain the 3D image of the whole sample, the focused beam scans the sample in the horizontal plane (xy-directions) and then mechanical refocuses at different depths (z-direction) in the sample. Afterward there is repeating the horizontal scanning which produces another thin ‘optical slice’.

### 3.6. Ultra-Violet-Visible Spectroscopy—Free Droplets

Measurement of the helix pitch was performed on a free droplet on a microscope slide using a spectroscopic method for wavelengths in the UV-VIS range (Liga microspectrometer, microParts GmbH, Dortmund, Germany). The sample’s image was registered during very slow cooling from the isotropic phase, using the Linkam hot stage and temperature stabilizer. The cooling rate was very slow, with a step of 0.1 °C, and required a one-minute isothermal time before each measurement.

Due to the fact the measurements were made on a free droplet, without any alignment, the raw data in the form of transmission spectra are only used for qualitative interpretation—determination of the minimum transmission λ_s_ [95]. Since we do not know the geometric dimensions of the droplets, the absorbance (transmission) at a given wavelength and temperature can’t be compared. However, different droplet thicknesses for Composites 1–3 do not affect the position of the maximum absorbance (minimum transmission).

### 3.7. Optical Microscopy/Scanning Electron Microscopy

Keyence VHX-7000 digital optical microscope using a built-in high-intensity LED light source was used to obtain the images of Composites 1–3 and pure Au nanoparticles. Each sample was mechanically dispersed with a metal roller on the surface of a chemically cleaned microscopic slide, obtaining a sample in the form of a square a few millimeters in size and a thickness up to ca. 100 μm. Each sample was imaged in two ways: as a pristine sample at room temperature and after annealing at 155 °C for 5 min and cooled down to room temperature. Digital optical images with high resolution were obtained by two different types of illumination: ring and coaxial. The first one gives the knowledge about the surface sample’s color simultaneously with no appropriate depth of contrast. This illumination system was just to confirm (at high magnification) a real sample color. In turn, the coaxial system is associated with perfect image depth, and it permits the recognition of details.

Scanning electron microscopy imaging was performed with a Keyence VHX-D510 scanning electron microscope (SEM)—(tungsten TE gun, acceleration voltage 0.9 kV, backscattered electron imaging) for Composites 1–3 and pure Au nanoparticles mounted to the SEM specimen stubs with the double-sided adhesive copper tape without an additional sputter-coating with metal to preserve original color and morphology of materials for the comparative analysis. The scanning electron microscope was the tandem system that allows obtaining both SEM and optical images. The maximum magnification in optical system imaging is 200×. Additionally, there is an option of overlaying SEM and optical images to form the superimposed image, which is very helpful to obtain additional information.

### 3.8. Contact Angle Measurements

All contact angle measurements were performed on a DSA20E EasyDrop wettability system (KRŰSS, Hamburg, Germany). The studied composites were dissolved in toluene (the same concentrations for all samples) and then spin-casted (single shot 100 μL, 820 rpm, 40 s) on the silicon wafers. The silicon wafers (ca. 1.00 × 2.00 cm) were previously cleaned in isopropanol (with the ultrasonic bath addition) and purged with high purity compressed nitrogen. The layers were subjected to aging for about 24 h to evaporate any residual solvent and the contact angle measurements were carried out during heating in the range of 0–100 °C. The wetting agent was distilled water, dosed dropwise per layer by a syringe. Measurement points were collected every 3 °C and the contact angle for each droplet was determined from its left and right side using DropAnalyzer software (KRŰSS, Hamburg, Germany). Both angles were then averaged at a given temperature. After the measurement, the droplet was gently collected with a paper towel, in a non-contact manner on the studied surface. The accuracy of temperature was ±0.1 °C, while the contact angle was subject to a systematic uncertainty of 0.1°. Measurements were made only up to 100 °C because the wetting agent used evaporates around 100 °C, so the measurements include crystalline phases and part of the antiferroelectric SmC*_A_ phase. Our goal was to determine whether the Au nanoparticles affect the surface wettability of prepared composites. A pure silicon wafer was used as a reference sample, the wettability of which was also studied under the same conditions as the composites. Due to the water solubility of the Au nanoparticles decorated layer, it was not possible to make CA measurements for Au NPs.

## 4. Conclusions

Properties of two composites based on antiferroelectric liquid crystal (S)-MHPOBC doped with two concentrations of Au nanoparticles: Composite 2 (0.2 wt. % Au NPs) and Composite 3 (0.5 wt. % Au NPs) were studied and compared with undoped (S)-MHPOBC (Composite 1). Based on the complementary methods, it was shown that many parameters are changed by a small amount of Au NPs.

Textures of the SmA*, SmC*_α_, SmC*, SmC*_γ_, and SmC*_A_ phases were not changed after doping, which may indicate the good dispersion of Au NPs in the LC matrix. However, due to a large number of focal conic defects, Au NPs may accumulate in these defects, reducing the free energy of the layer. In turn, a wide temperature range of the SmI* and Cr_1_ phases coexists (as a result of crystallization inhibition) and a new Cr_2′_ crystal phase was revealed after doping.

Doping has a very slight influence on the phase sequence and transition temperatures; only additional crystal phases Cr_4′_ and Cr_2′_ were revealed for Composite 3 during heating. During heating, there was an increase of ΔH for the Cr_4_–SmC*_A_ phase transition, as well as for transitions between sub-phases, while there was a decrease of ΔH for the SmA*–Is transition both during heating and cooling that was observed after doping.

The smectic layer thickness does not change after doping in the uncertainty limit; only a slight modification of the correlation length in the SmA* phase for Composite 2 was observed, which is evidence of the influence of Au nanoparticles on the ordering of LC molecules within the smectic layer. In turn, the increase of the helix pitch after doping with Au NPs was observed in the SmC*_A_ phase, especially for Composite 3.

The surface morphology does not change after doping; however, after heating up to the isotropic phase (150 °C) and cooling down black objects, white spots appeared on the surfaces of Composites 2 and 3, which means that, in the isotropic liquid, the nanoparticles aggregate. In turn, pillar-like structures for pure Au nanoparticles were observed, which disappear when heated up to 150 °C. This is most likely related to the destruction of the lamellar structure formed by decorated nanoparticles at higher temperatures. The contact angle changed after doping, and it was the highest for Composite 2 in the crystal phase and Composite 3 in the SmC*_A_ phase. The differences in Θ_CA_ reach even 15–17° compared to undoped (S)-MHPOBC (Composite 1), which means that the surface wettability is reduced. This may be the result of conformational changes in the surfactant chains decorated with Au NPs. Both in the range of green and blue light, we observed an increase in the fluorescence intensity after doping with Au nanoparticles, while, for the green light range, the gain is almost 100%.

The switching time decreased for Composite 3 and increased for Composite 2 in the antiferroelectric SmC*_A_ phase. In turn, the tilt angle increased after doping in Composite 3, while it did not change in Composite 2 in the entire SmC*_A_ phase. Spontaneous polarization increased after doping at the high temperature range of the SmC*_A_ phase, while, below ca. 90 °C, no dopant influence was observed. The increase in P_s_ resulted from the induction of the dipole moment of Au NPs under the influence of the incident light.

In summary, we found that the admixture of gold nanoparticles with low concentrations affects the fluorescence and hydrophobicity of the composite layers, as well as the parameters of the SmC*_A_ phase, such as the helix pitch, switching time, tilt angle, and spontaneous polarization, but an increase in the dopant concentration led to NP agglomeration. In turn, the admixture of Au NPs did not affect the phase sequence, thickness of the smectic layers, and correlation length in the SmC*_A_ phase, as well as the helix pitch, switching time, tilt angle, and spontaneous polarization in the SmI* phase.

However, there is still a lot of research to be done. First of all, based on the (S)-MHPOBC, new composites can be prepared and investigated whether the spontaneous polarization, contact angle, tilt angle, and enthalpy change for the Cr_4_–SmC*_A_ phase transition will increase with the increasing concentration of Au NPs. It is also possible to check the influence of the Au nanoparticle sizes and the type of surfactants decorating them on these parameters and whether other antiferroelectric liquid crystals as the host matrix provide the same modifications as these parameters. It is also possible to investigate the effect of Au NP: (1) on the relaxation processes in the LC matrix by dielectric spectroscopy, (2) on the absorption bands by infrared spectroscopy, (3) on the maximum luminescence (hypsochrome or bathochrome shift) by means of photoluminescence measurements, and (4) on electronic transitions in the LC matrix by UV–Vis spectroscopy of thin layers.

The composites studied in this work are interesting from the point of view of applications as, for example, electric switches or for the production of thin hydrophobic films protecting against moisture. In turn, due to the fact that both the LC matrix and the composites show fluorescence in the visible range and are nontoxic, such materials can be used as dyes/markers for medical applications.

## Figures and Tables

**Figure 1 molecules-27-03663-f001:**
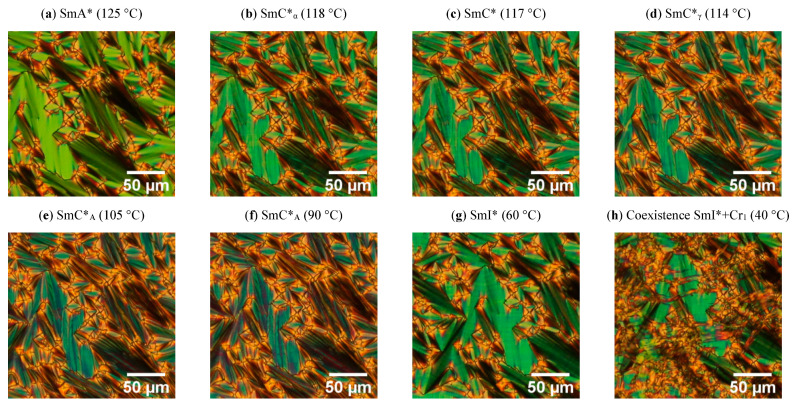
Textures under cross-polarizers registered during cooling with a rate of 2 °C/min for Composite 3: SmA* (**a**), SmC*_α_ (**b**), SmC* (**c**), SmC*_γ_ (**d**), SmC*_A_ at 105 °C (**e**), SmC*_A_ at 90 °C (**f**), SmI* (**g**), and the coexistence of two phases, SmI* and Cr_1_ (**h**). All images present the same sample area.

**Figure 2 molecules-27-03663-f002:**
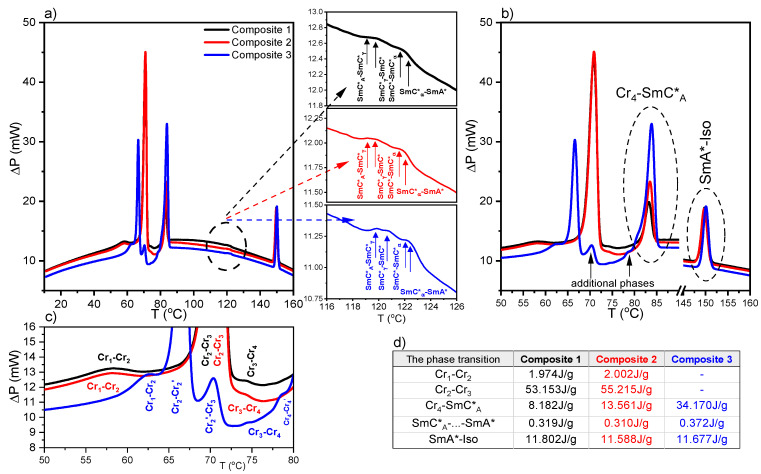
Calorimetric curves registered during heating for Composites 1–3 (**a**). Insets in (**a**) represent an enlarged temperature range of 116–126 °C for individual Composites. Enlarged temperature range of the DSC curves registered for Composites 1–3 with additional phase transition for Composite 3 marked by a vertical arrow (**b**), DSC curves at low temperature range (**c**), and table with the enthalpy changes for the chosen phase transitions (**d**).

**Figure 3 molecules-27-03663-f003:**
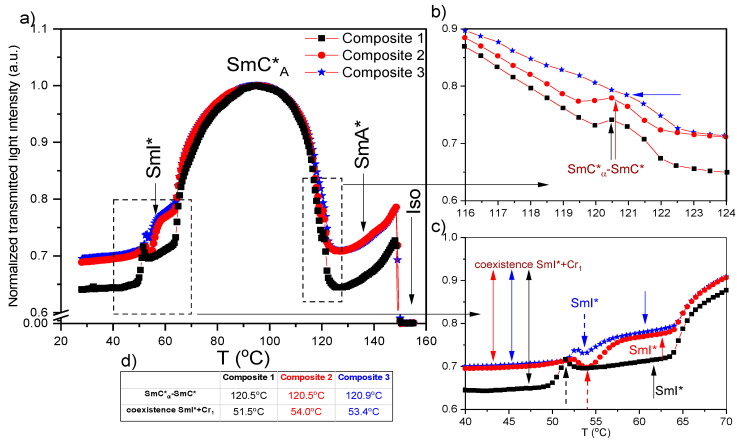
Temperature dependence of normalized transmitted light intensity during cooling (**a**), the selected temperature range in the vicinity of the SmC*_α_–SmC* (**b**), and the SmC*_A_–SmI*–Cr_1_ phase transition (**c**), determined phase transition temperature for SmC*_α_–SmC*, and temperatures at which are observed the coexistence regime (**d**); the accuracy of the temperature determination is equal to ±0.1 °C.

**Figure 4 molecules-27-03663-f004:**
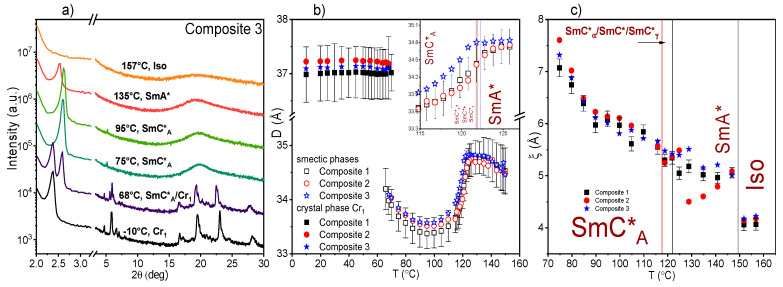
Diffraction patterns at several chosen temperatures as an example for Composite 3 (**a**), the temperature dependence of the smectic layer spacing/interplanar distance in a crystal phase (**b**), and the correlation length (**c**) on cooling for all Composites. The inset in (**b**) shows the high temperature range with colored lines representing the SmA*–SmC*_α_ phase transition temperature obtained by DSC method for all Composites.

**Figure 5 molecules-27-03663-f005:**
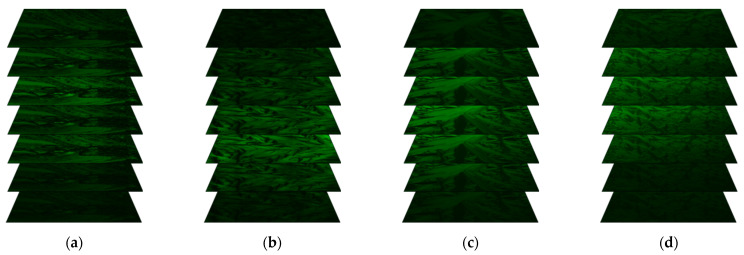
Fluorescence images in subsequent slices for the Au nanoparticles (at room temperature) (**a**), Composite 1 (**b**), Composite 2 (**c**), and Composite 3 (**d**) registered in the spectral ranges 490–590 nm. For Composites 1–3, the images were registered at 68 °C (SmC*_A_).

**Figure 6 molecules-27-03663-f006:**
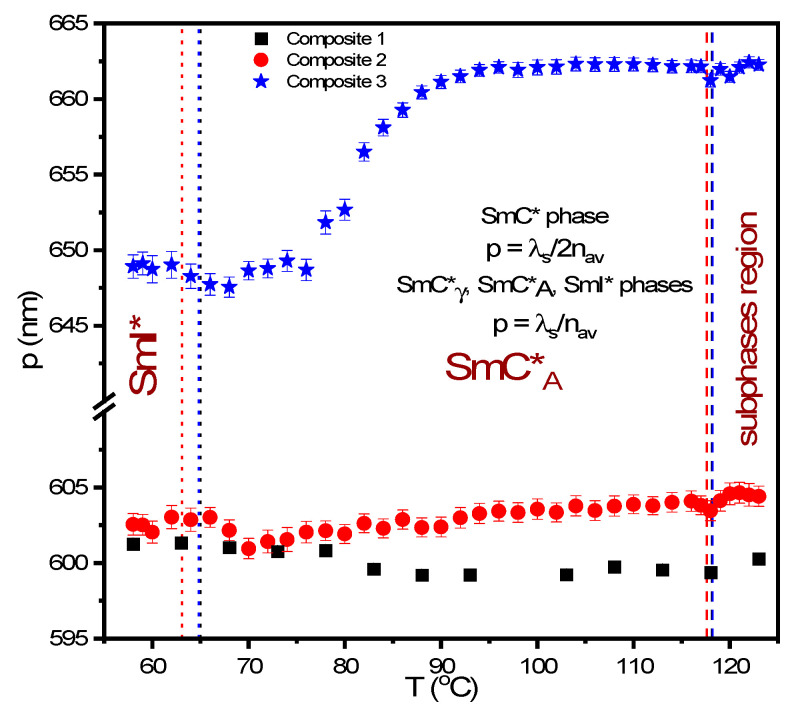
Temperature dependence of the helix pitch for Composites 1–3 obtained during cooling together with the relations between the helix pitch and the wavelength in the maximum transmission for the antiferroelectric and ferroelectric phases. The dashed and dotted vertical lines indicate the SmC*_γ_–SmC*_A_ and SmC*_A_–SmI* phase transition temperatures obtained by DSC.

**Figure 9 molecules-27-03663-f009:**
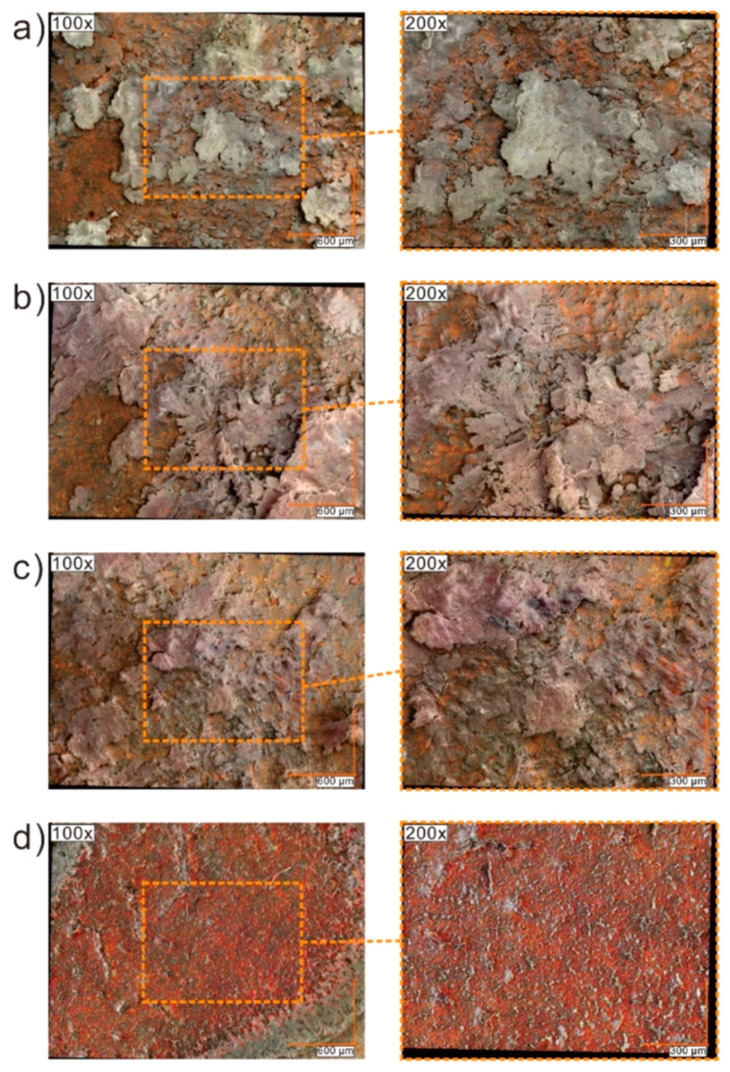
Real colored SEM images for Composite 1 (**a**), Composite 2 (**b**), Composite 3 (**c**), and Au nanoparticles (**d**) obtained by superimposing the SEM images on the optical images.

**Figure 10 molecules-27-03663-f010:**
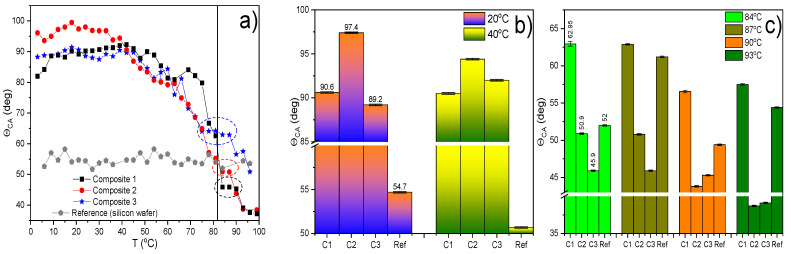
Temperature dependence of the contact angle for Composites 1–3 (C1, C2, and C3), and silicon wafer (Ref) during heating from 0 °C up to 100 °C (**a**). The dashed ellipses outline the step related to the Cr–SmC*_A_ phase transition. Bar chart of contact angle in the crystal phase for two chosen temperatures (**b**) and in the SmC*_A_ phase for the four chosen temperatures (**c**). Error bar is a sum of systematic uncertainty and standard deviation for left and right drop contact angles.

**Figure 11 molecules-27-03663-f011:**
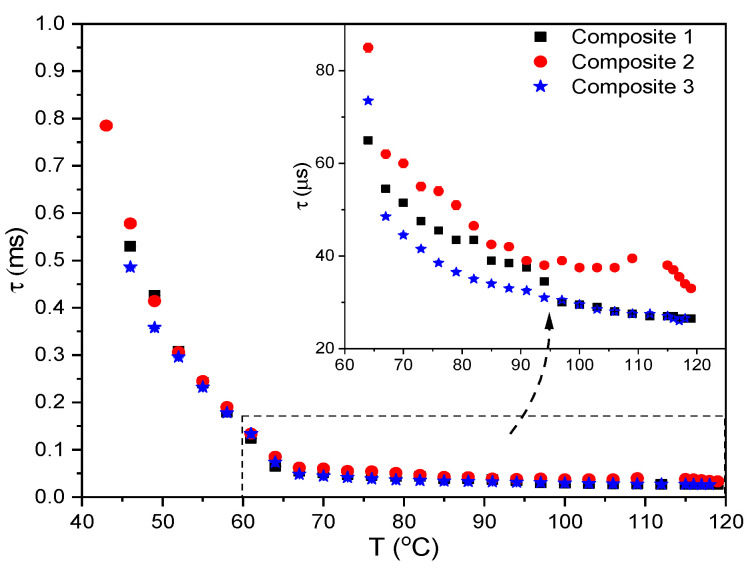
Temperature dependence of the switching time, f = 50 Hz, U = 120 V. The rectangular dashed area presents the SmC*_A_ range at the inset.

**Figure 12 molecules-27-03663-f012:**
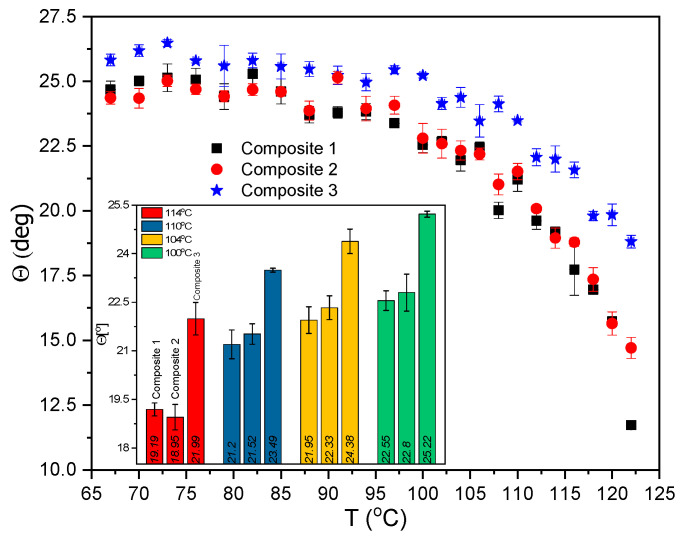
Temperature dependence of the tilt angle for Composites 1–3, f = 50 Hz, U = 120 V within the SmC*_A_ phase. Inset shows the tilt angle for the chosen temperatures.

**Figure 13 molecules-27-03663-f013:**
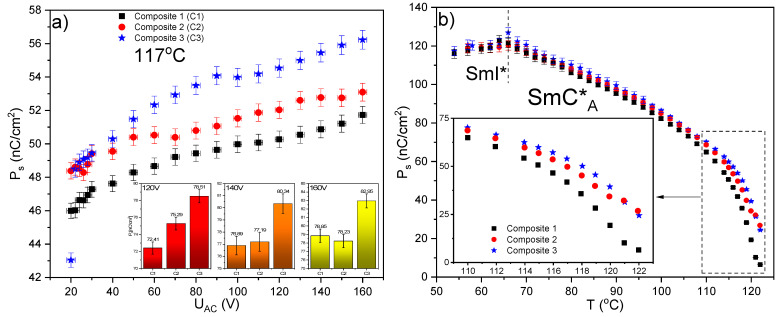
Spontaneous polarization versus applied voltage, T = 117 °C, f = 50 Hz (**a**). The inset shows the P_s_ bar graphs for the selected amplitudes at T = 105 °C, f = 50 Hz. Temperature dependence of the spontaneous polarization in the SmC*_A_ and SmI* phases for all Composites (**b**). An enlarged high temperature range is presented in the inset, ΔU_AC_ = ±2 V, ΔP_s_ = 1%.

**Figure 14 molecules-27-03663-f014:**
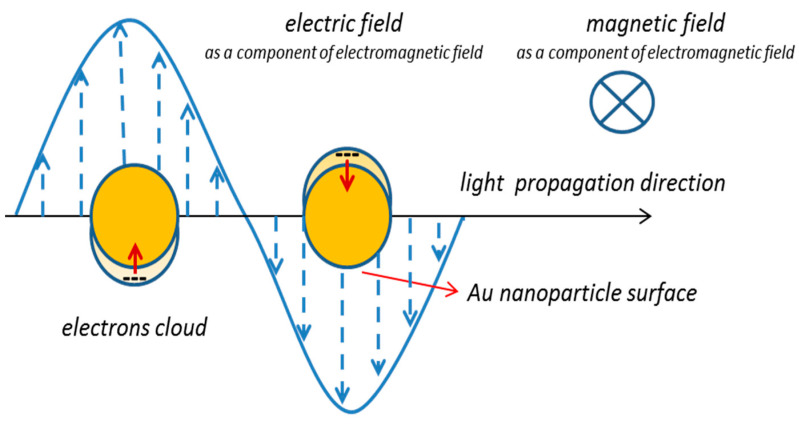
Schematic illustration of the localized surface plasmon resonance for spherical Au nanoparticles. Red arrows mean an induced dipole moment, which is the result of displacement electrons cloud (marked by “---“). The dashed arrows mean electric field lines, and the orange circle represents an Au nanoparticle.

**Figure 15 molecules-27-03663-f015:**
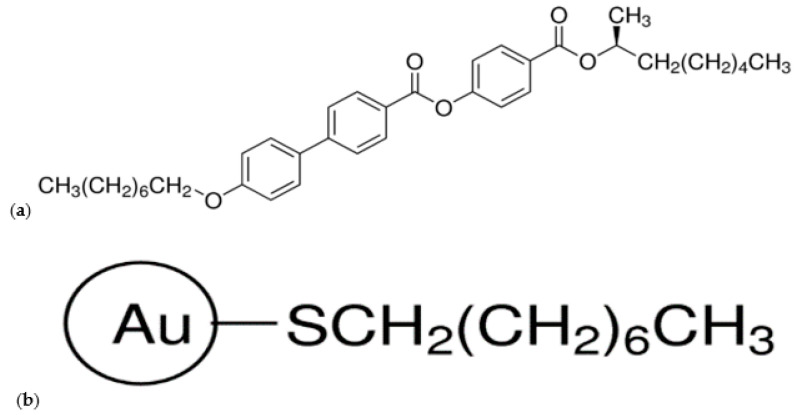
Chemical structure of liquid crystal (S)-MHPOBC (**a**) and the 1-octanethiol functionalized gold nanoparticle (**b**).

**Figure 16 molecules-27-03663-f016:**
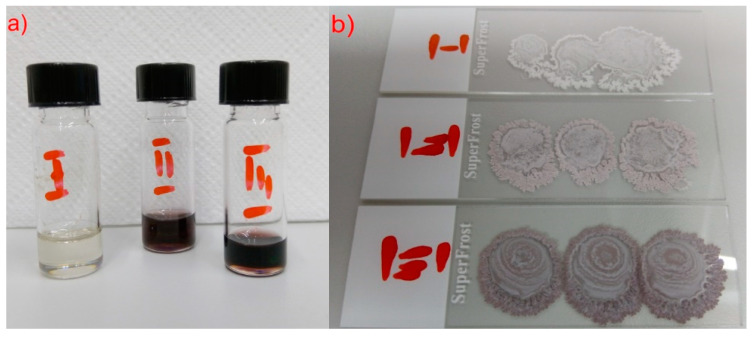
Composites as solutions (in 2ml toluene) (**a**) and after evaporation on the microscopic slides (**b**): I—Composite 1, II—Composite 2, and III—Composite 3.

**Table 2 molecules-27-03663-t002:** Mass and volume compositions of pure (S)-MHPOBC and nanoparticle solutions in toluene of Composites 1–3.

**Composite**	**(S)-MHPOBC**	**Au NPs**
Composite 1 (0.0 wt. %)	100.03 mg	-
Composite 2 (0.2 wt. %)	100.02 mg	10.02 μL
Composite 3 (0.5 wt. %)	60.00 mg	15.08 μL

## Data Availability

Not applicable.

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
