# Peer review of "Nanocomposites Based on Antiferroelectric Liquid Crystal (S)-MHPOBC Doping with Au Nanoparticles"

_molecules, 2022, doi:10.3390/molecules27123663_

Round 1

Reviewer 1 Report

Here are a few questions and suggestions to improve the manuscript:

(1) Please make sure that all abbreviations are correctly introduced when unsed for the first time; e.g. CB, MBBA, CHBT, POM, DSC, etc.

(2) In the introduction, the stated goal is to give a brief overview of the research of nanocomposites based on liquid crystals and Au nanoparticles to demonstrate the need for the research and explain the motivation. However, this overview goes into great detail. It would be helpful to give a more general overview in the introduction in order to reach a wider audience, instead of giving to much details at this stage, including numerous approaches of other researchers.

(3) In line 357 it is stated "for the Composite 2 in the SmA* phase where little decrease with temperature is visible (Fig. 4c)" - However, Figure 4 shows only Diffraction patterns for Composite 3. ?

(4) Please make sure that all details are given in the Methods Section that are needed to repeat the experiments.

(5) What is lacking is an outlook at the end that places the significance of the research in a larger context and addresses future steps or research questions.

Reviewer 2 Report

Paper by Dr. Monika Marzec et al “Nanocomposites based on antiferroelectric liquid crystal (S)- 2 MHPOBC doping with Au nanoparticles” describes composites based on antiferroelectric liquid crystal doped with Au nanoparticles. Authors used a number of experimental techniques to study comprehensively different properties of the composites in order to elucidate the influence of nanoparticles on LC properties. The presented results are quite interesting but in most cases they need more careful explanations and interpretations.

  1. Page 4, line 156. Composite 1 does not contain any additive, thus, in order to avoid misunderstanding better to name it as (S)-MHPOBC.
  2. Page 7, bottom. The information about cooling rate should be added.
  3. Page 8, Fig. 3d. The error in temperature determination should be added.
  4. Page 10, line 382. “We can’t unambiguously state whether this fluorescence is coming from Au nanoparticles or/and surfactant, because Au nanoparticles used were decorated with 1-octanethiol”. 1-octanethiol does not have fluorescence due to the lack of appropriate chromophores fragment. The overall description of the fluorescent properties of the composites is unsatisfactory; there is no adequate discussion on this point. Images in Fig. 5 do not show any significant difference; at least authors did not provide full description of those.
  5. Page 11, Figs. 6b, c and related text. “The maximum transmission along with the uncertainty was appointed based on fitting the Gaussian curve to the experimental data…” Selective reflection of chiral helical LC (cholesteric or SmC*) appears in transmittance spectra as minimum, rather than maximum. Explanation of this contradiction should be added.
  6. Page 10, lines 411-413. “The temperature dependence of the helix pitch for all Composites is similar and qualitatively the same as that observed for (R)-MHPOBC by Chandani et al. [58].“ As seen from Fig. 6a character of temperature dependence and values of pitch are completely different.
  7. Page 12, line 454. The formation of “one-dimensional lamellar structure” should be supported by other methods, such as X-rays.
  8. Page 14, line 528. Plateau mentioned in the text is hardly visible in Fig. 10a and, probably, its existence is under big question due to chaotic character of experimental curves.
  9. Page 17, lines 607-608. “The sample of Composites were exposed to white light during the electro-optic measurements which may be responsible for the generation of localized surface plasmon resonance.” Information of light intensity and, especially, the influence of light intensity on the Ps values must be experimentally measured and analyzed.

               Summarizing, this manuscript can be published only after major revision.

Reviewer 3 Report

The (S)-MHPOBC antiferroelectric liquid crystal (AFLC) doped with low concentrations of gold nanoparticles was studied in this manuscript. Several experimental methods were used to determine the effect of Au nanoparticles on AFLC in the metal-organic composites. (i) Authors used polarizing microscopy, observing that all composites studied show a rich phase polymorphism. In general, the addition of Au nanoparticles did not change the texture of the phases. (ii) DSC measurements present a series of phase transitions. Authors observed a small concentration of Au NPs strongly influences the transition between crystal and liquid crystalline phases. Transmitted light intensities versus temperature during cooling was used to complement the other techniques for some samples. (iii) TLI and XRD methods confirm the existence of a hexatic phase below the SmC*A one. The LC matrix and Au nanoparticles show strong fluorescence in the green light range. (iv) High resolution optical images suggest the movement of Au nanoparticles in the liquid crystal matrix (in the isotropic phase), leading to their agglomeration. Authors reported for some samples that low concentration of Au NPs does not disturb the original liquid crystal order. Authors also studied spontaneous polarization, observing results explained with help of the literature and previous works. I think the manuscript can be published after some amendments. Please, see below: 

1. Authors say that "The SmC*-SmC*α phase transition is not visible, although it was observed by Chandani et al. for (R)-MHPOBC at a very low heating rate [58]." Is this expected, I mean, most of the times when you decrease the heating rate you find it difficult to observe a phase transition. Please, explain it.

2. "The greatest impact of the Au NPs on the enthalpy changes was found for the Cr4-SmC*A phase transition in Composite 3" The difference of enthalpy between composite 1 and 3 is very high; is this expected?

3. The blue arrow in Fig. 3(b) points to nothing. There is no modification where the arrow is pointing.

4. "We can’t unambiguously state whether this fluorescence is coming 382 from Au  nanoparticles or/and surfactant, because Au nanoparticles used were decorated 383 with 1-octanethiol." This doubt should be easily eliminated if authors had used, for example, Raman spectroscopy or other techniques to observe the vibrations (or not) of 1-octanethiol. Why not seek to understand this point?

5. I suggest improving Figs. 6(b) and (c); change the size of numbers in the horizontal and vertical scale (increasing the interval between the numbers, for example, 400, 500, 600 nm, etc). 

Round 2

Reviewer 2 Report

The authors tried to performed necessary corrections but the paper still have to be revised because several points are not clear yet.

1.       Fig. 6a. Author should explain difference between pitch values for composite 2 and 3 and discuss its origin in the text.

2.       Fig. 6c. Authors changed transmittance simply by changing letter “T” to letter “A” and according to this figure absorbance reaches almost 90%. According to these values, material selectively reflects both circular polarizations of light. Authors should add explanation of this extraordinary phenomenon.
